Journal of
**open** psychology data

# Using the Probability-Based GESIS Panel for Longitudinal Psychological Research on the COVID-19 Outbreak in Germany

**DATA PAPER**

**BERND WEIß** ⬤

**SVEN STADTMÜLLER** ⬤

*Author affiliations can be found in the back matter of this article

## ABSTRACT

The GESIS Panel (GP) allows the scientific community to collect survey data free of charge within a probability-based mixed-mode panel. In addition, the GP provides high-quality longitudinal data for secondary research. In March 2020, the GP conducted a special survey on the coronavirus outbreak. Since then, the COVID-19 questionnaire module has become an integral part of the GP. In sum, 13 waves were fielded that included questions on COVID-19, e.g., on attitudes, opinions, and behaviors related to the pandemic. Here, we introduce the GP focusing on longitudinal psychological research opportunities regarding the COVID-19 outbreak.

**CORRESPONDING AUTHOR:**
**Bernd Weiß**

GESIS – Leibniz Institute for the Social Sciences, Germany

bernd.weiss@gesis.org

---

**KEYWORDS:**
Survey; longitudinal data; COVID-19; pandemic; attitudes and behaviors

**TO CITE THIS ARTICLE:**
Weiß, B., & Stadtmüller, S. (2023). Using the Probability-Based GESIS Panel for Longitudinal Psychological Research on the COVID-19 Outbreak in Germany. *Journal of Open Psychology Data,* 11: 16, pp. 1–12. DOI: https://doi.org/10.5334/jopd.90

# (1) BACKGROUND

The COVID-19 pandemic had (and still has) an immense impact at the societal and individual level in Germany. During the pandemic, metrics like the incidence, vaccination, or hospitalization rate were highly important. In a similar vein, the consequences of the pandemic are mostly summarized with aggregated data like the number of deaths or economic metrics (e.g., the growth rate and the number of insolvencies). Since the pandemic, despite all regional differences, had entire Germany firmly in its grip, this orientation towards aggregated data is reasonable. However, data tracking individual attitudes, behaviors, and mental health over the course of the pandemic is also essential for several reasons. First, the pandemic affected people's lives differently. For instance, parents of young children, people working in jobs belonging to the 'critical infrastructure', older people, and people with pre-existing health conditions were particularly challenged by the pandemic. Thus, microdata allow estimating the pandemic's consequences for different population groups. Second, the challenges of the pandemic varied between individuals and *over time* as the pandemic proved to evolve highly volatile. In this sense, it is important to learn how people coped with the varying measures during the pandemic, as this can help in developing strategies to promote individual and collective resilience. Lastly, policymakers also depend on individual data since any political intervention necessary to prevent the further spread of COVID-19 infection entails far-reaching restrictions for society. Moreover, the success of the restrictions depends on citizens' compliance (Schaurer & Weiß, 2020).

To meet these demands, the open[1] probability-based German GESIS Panel (GP) started its data collection utilizing an online subsample early in March 2020, i.e., only six weeks after the first case of COVID-19 in Germany was confirmed. The data of the so-called "GESIS Panel Special Survey on the Coronavirus SARS-CoV-2 Outbreak in Germany" was then published in April 2020. Since then, a questionnaire module on the COVID-19 outbreak has become an integral part of the GP from May 2020 onwards. By the time of publishing, 13 survey waves were fielded that include questions on COVID-19 (for an overview of all waves, see Table 1). More specifically, the COVID-19 module covers attitudes, opinions, and behaviors related to the pandemic, such as the assessment of the risk of infection, adopted measures to prevent infection, attitudes and behaviors toward vaccination, trust in politics and institutions, and psychological responses (e.g., mental health) to the pandemic.

In this paper, we introduce the GP, focusing on longitudinal psychological research opportunities regarding the COVID-19 outbreak. We start by providing an overview of the GP, its study and sampling design, and a more in-depth description of the COVID-19 module. We also highlight some instruments that we deem relevant for personality-related research. We conclude with some ideas illustrating the reuse potential of the GP, and the COVID-19 module in particular, as well as a longitudinal use case for psychological research using data on the reported intention to vaccinate against COVID-19.

# (2) METHODS

In this section, we will introduce the GP, a permanent longitudinal survey infrastructure in Germany, as well as the "GESIS Panel Special Survey on the Coronavirus SARS-CoV-2 Outbreak in Germany" (GP Corona). The latter one was conducted in addition to the regular GP waves. Many of the COVID-19 pandemic-related instruments that we introduced in the GP Corona survey were also fielded later on in regular waves of the GP, enabling users to run sophisticated longitudinal analyses. When we describe the GP data, we refer to the most recent published version of the GP, wave jb, which was fielded between May and July 2022 (GESIS, 2023a, 2023b; Minderop et al., 2023).[2]

## 2.1 STUDY DESIGN

The GP is a panel infrastructure initiated in 2013 and operated by GESIS – Leibniz Institute for the Social Sciences in Mannheim, Germany (Bosnjak et al., 2018). The GP is a probability-based survey. This means that the selection of participants in Germany is based on a random sample of individuals from the German-speaking residential population. Moreover, the GP is a mixed-mode survey as data collection is carried out by web and mail. Hereby, the net sample of the GP also includes those parts of the population that do not have Internet access or are not willing or sufficiently able to answer the questionnaire online (Bretschi & Weiß, 2023). About 75% of the respondents participate online (Computer-Assisted Web Interview, CAWI), and 25% of our sample prefer a paper-based questionnaire (Paper and Pencil Interview, PAPI) sent by (postal) mail.

One unique selling point of the GP – in addition to using its data for secondary research – is that any researcher can submit a primary research proposal to collect survey data within the panel. The proposals undergo external peer review and are evaluated based on scientific and survey methodological quality.[3] That is, the GP is an omnibus survey covering a broad range of topics from various disciplines. Therefore, the GP is also highly suited as a data source for secondary data analysis when it comes to scales and instruments that are less mainstream but are collected within a large-scale probability-based sample.

The initial GP Corona (GESIS Panel Team, 2020) was conducted between March 17, 2020, and March 29, 2020.

Since the survey data had to be collected and published in a timely manner, only the online subsample of GP respondents was invited to participate. From May 2020 onwards, the questionnaire module on the COVID-19 outbreak has become an integral part of the GP, i.e., utilizing the online as well as the offline sample.

Until now (August 2023), one special survey and 13 panel waves that included COVID-19 items were fielded. The special survey (study hz) and ten of these 13 waves have already been published, i.e., the regular survey waves hb to jb (see Table 1). Each new data release contains the cumulative data set from all previous survey waves as well as from GP Corona. The most recent data publication that can be used for data analyses is version 48 (GESIS, 2023a, 2023b). The recommended way to work with all GP COVID-19 instruments is to obtain the most recent GESIS Panel Standard/Extended Edition (see Section (3) for a more detailed explanation of these two editions). Accordingly, we refer to this version when we report on GP data.

Since the measures to contain the pandemic varied over time, we have updated the questionnaire module on the COVID-19 outbreak for the last three years. For instance, in the first survey waves, we asked what respondents thought of a curfew and whether they would comply with it. In later waves, we included questions on the evaluation of various access regulations, masking obligations, and attitudes and behaviors toward vaccination. However, many constructs and the associated items have been fielded (almost) over the entire period, including questions on the self-assessed risk of infection, adopted measures to prevent infection, support for public measures, trust in politics and institutions, and mental health (this module has study code cy, and can be found in detail in GESIS, 2023c; see also Table 1 for an overview). In addition, we fielded a number of external fast-track/short-submission studies that also address various COVID-19-related research questions, such as "Resilience in the Corona Crisis", "The

Corona-Conspiracy", "Housing and Partnerships during the Corona Lockdown", or "Understanding the Corona-Warn-App Download and Use" (GESIS, 2023c).

## 2.2 SAMPLE AND DATA COLLECTION

Initially, the GP targeted the German-speaking population aged between 18 and 70 years (at the time of the initial recruitment in 2013) and permanently residing in Germany. Sampling for the GP was based on the population registers of randomly selected German municipalities (register-based sampling). However, starting with the first refreshment sample in 2016, the upper age limit has been discarded. Two more refreshment samples were drawn in 2018 and 2021; a fourth refreshment sample will be introduced into the GP in the first survey wave in 2024 (wave "la"). The sampling design of all refreshment samples was based on a "piggybacking" approach, i.e., respondents were recruited after completing other GESIS surveys. The first two refreshment cohorts (2016, 2018) were based on the ALLBUS, the German General Social Survey, while the third refreshment sample (2021) was drawn from the German part of the ISSP, the International Social Survey Programme. Currently, new panelists for the fourth refreshment cohort are recruited among participants of the German part of the European Social Survey (ESS).

The GP currently has about 5,200 respondents that are surveyed on a quarterly basis, i.e., four waves per year (until 2021, it used to be six waves per year). So, up until August 2023, 51 regular survey waves have been fielded since 2014 (excluding the three recruitment waves in 2013; the following website provides an overview of all waves: https://www.gesis.org/gesis-panel/gesis-panel-home/wave-schedule, accessed on 2023-08-17). For each survey wave, respondents receive 5 euros as an unconditional incentive.

To reduce mode measurement effects, the GP pursues a unified mode approach (Dillman & Edwards,

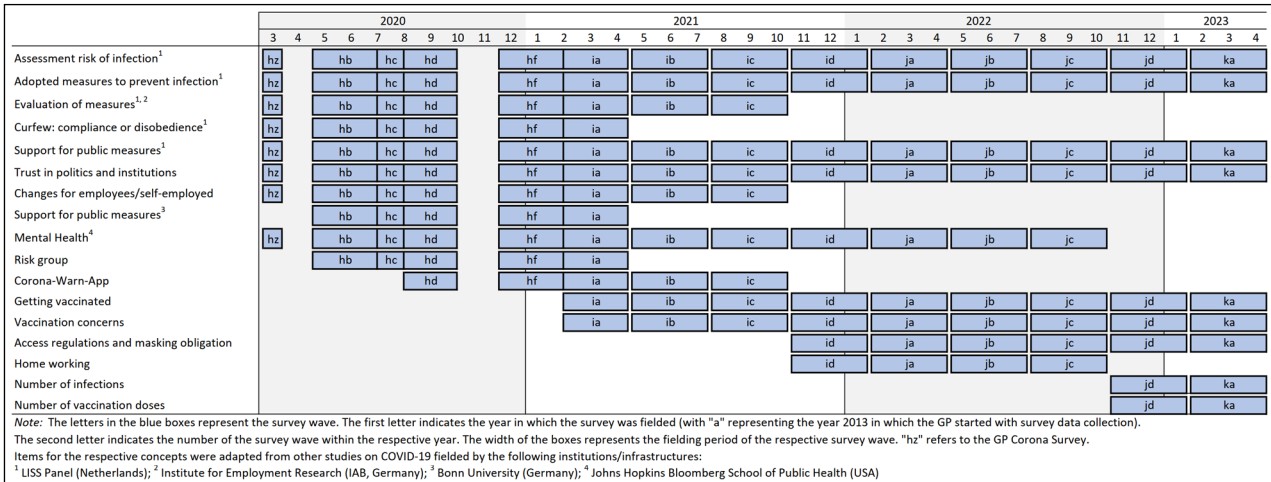

Years: 2020 covers months 3–12; 2021 covers months 1–12; 2022 covers months 1–12; 2023 covers months 1–4.

| | 3 | 4 | 5 | 6 | 7 | 8 | 9 | 10 | 11 | 12 | 1 | 2 | 3 | 4 | 5 | 6 | 7 | 8 | 9 | 10 | 11 | 12 | 1 | 2 | 3 | 4 | 5 | 6 | 7 | 8 | 9 | 10 | 11 | 12 | 1 | 2 | 3 | 4 |
|---|---|---|---|---|---|---|---|---|---|---|---|---|---|---|---|---|---|---|---|---|---|---|---|---|---|---|---|---|---|---|---|---|---|---|---|---|---|---|
| Assessment risk of infection[1] | | | hz | hb | | hc | | hd | | | hf | | ia | | ib | | | ic | | | id | | | ja | | | jb | | | jc | | | jd | | | ka | | |
| Adopted measures to prevent infection[1] | | | hz | hb | | hc | | hd | | | hf | | ia | | ib | | | ic | | | id | | | ja | | | jb | | | jc | | | jd | | | ka | | |
| Evaluation of measures[1,2] | | | hz | hb | | hc | | hd | | | hf | | ia | | ib | | | ic | | | | | | | | | | | | | | | | | | | | |
| Curfew: compliance or disobedience[1] | | | hz | hb | | hc | | hd | | | hf | | ia | | | | | | | | | | | | | | | | | | | | | | | | | |
| Support for public measures[1] | | | hz | hb | | hc | | hd | | | hf | | ia | | ib | | | ic | | | id | | | ja | | | jb | | | jc | | | jd | | | ka | | |
| Trust in politics and institutions | | | hz | hb | | hc | | hd | | | hf | | ia | | ib | | | ic | | | id | | | ja | | | jb | | | jc | | | jd | | | ka | | |
| Changes for employees/self-employed | | | hz | hb | | hc | | hd | | | hf | | ia | | ib | | | ic | | | | | | | | | | | | | | | | | | | | |
| Support for public measures[3] | | | | hb | | hc | | hd | | | hf | | ia | | | | | | | | | | | | | | | | | | | | | | | | | |
| Mental Health[4] | | | hz | hb | | hc | | hd | | | hf | | ia | | ib | | | ic | | | id | | | ja | | | jb | | | jc | | | | | | | | |
| Risk group | | | | hb | | hc | | hd | | | hf | | ia | | | | | | | | | | | | | | | | | | | | | | | | | |
| Corona-Warn-App | | | | | | | | hd | | | hf | | ia | | ib | | | ic | | | | | | | | | | | | | | | | | | | | |
| Getting vaccinated | | | | | | | | | | | | | ia | | ib | | | ic | | | id | | | ja | | | jb | | | jc | | | jd | | | ka | | |
| Vaccination concerns | | | | | | | | | | | | | ia | | ib | | | ic | | | id | | | ja | | | jb | | | jc | | | jd | | | ka | | |
| Access regulations and masking obligation | | | | | | | | | | | | | | | | | | | | | id | | | ja | | | jb | | | jc | | | jd | | | ka | | |
| Home working | | | | | | | | | | | | | | | | | | | | | id | | | ja | | | jb | | | jc | | | | | | | | |
| Number of infections | | | | | | | | | | | | | | | | | | | | | | | | | | | | | | | | | jd | | | ka | | |
| Number of vaccination doses | | | | | | | | | | | | | | | | | | | | | | | | | | | | | | | | | jd | | | ka | | |

*Note:* The letters in the blue boxes represent the survey wave. The first letter indicates the year in which the survey was fielded (with "a" representing the year 2013 in which the GP started with survey data collection).
The second letter indicates the number of the survey wave within the respective year. The width of the boxes represents the fielding period of the respective survey wave. "hz" refers to the GP Corona Survey.
Items for the respective concepts were adapted from other studies on COVID-19 fielded by the following institutions/infrastructures:
[1] LISS Panel (Netherlands); [2] Institute for Employment Research (IAB, Germany); [3] Bonn University (Germany); [4] Johns Hopkins Bloomberg School of Public Health (USA)

**Table 1** Overview of GESIS Panel waves that include items or instruments about the COVID-19 outbreak.

2016). This means that questions and response options between the two modes of data collection (i.e., web and mail) are displayed as similarly as possible. To account for the increasing share of panelists taking part with mobile devices, we also converted the questionnaire into a mobile-first layout in 2020. This means that the questions and response options are also displayed similarly between the different devices that can be used for web survey participation (i.e., desktop/laptop, tablet, and smartphone devices). Specifically, in the web questionnaire all questions and scales are now arranged vertically with up to five survey items per page. Consequently, the paper questionnaire was also rearranged from a single- to a two-column layout to account for the additional space the new layout required (Schwerdtfeger et al., 2022).

As introduced earlier, all pandemic-related data collections started with the special survey (GP Corona), which includes a wide range of questions on attitudes and behaviors regarding COVID-19 (see above). The target population was confined to the online subsample of the GP to speed up data publication. A total of 3,765 panelists were invited to participate, of whom 3,176 completed the survey, resulting in a completion rate of 84.36%. The GP Corona has been linked to the GESIS Panel Standard Edition (GESIS, 2023b) as well as the GESIS Panel Extended Edition (GESIS, 2023a) (see Section (3) for a more detailed explanation of these two editions). The web and mail sample differ largely in terms of socio-demographics: While in wave jb the mean age of online participants was 54.5 years, panelists who took part with the paper questionnaire were, on average, remarkably older (65.6 years). Moreover, the share of online panelists who obtained the highest upper secondary leaving certificate was 55.7% while only 22.8% for the offline panelists. Hence, data from GP Corona and from the regular panel waves should not be used together for analyzing changes over time (i.e., treating them as time series data or if they came from multiple cross-sectional surveys). If this is of key interest, we advise to only relying on the online subsamples.

As of August 2023, the most recent published data from the GP is wave jb. For this wave, 5,218 active panelists had been invited, of which 4,786 (completion rate: 91.72%) fully or partially completed the survey (Minderop et al., 2023). Table 2 depicts the respective unweighted and weighted sample composition with respect to gender, age, and educational attainment. The GP team provides design weights since the integration of the second, third and fourth cohort of the GP requires the calculation of design weights due to different inclusion probabilities of the respondents (more information on weighting can be found in Kolb et al., 2022, p. 9). According to Table 2, the average respondent is around 57 years old. While at first sight, the sample seems rather old, it should be highlighted that our target population consists of all German-speaking individuals 18 years and older. The German Federal Statistical Office provides a broad overview of the age distribution (in 2021) in Germany (Statistisches Bundesamt, 2023). Based on this information, we calculated an approximate average age of 54.43 years. Similar findings can be reported for gender, i.e., the distribution of gender in the sample (50% female) corresponds to that in the adult population in Germany (51% female; see Statistisches Bundesamt, 2020). With respect to educational attainment, though, we observed an over-representation of highly educated respondents: almost 50% of the respondents had obtained the highest upper secondary leaving certificate, that is, a "general higher education entrance qualification", compared with about 33% of the adult population in Germany (Statistisches Bundesamt, 2019, p. 88).[4] This over-representation of highly educated respondents comes as no surprise, and it is in line with findings on the demographic composition of online surveys (AAPOR et al., 2010; Antoun, 2015).

## 2.3 MATERIALS/SURVEY INSTRUMENTS

The GP Corona and subsequent GP waves included a wide range of questions on attitudes and behaviors regarding COVID-19. The initial COVID-19 GP questionnaire was designed in the context of the Open Probability-based

| | UNWEIGHTED | | | | | WEIGHTED | | |
|---|---|---|---|---|---|---|---|---|
| | n | M | SD | min | max | n | M | SD |
| Gender: Female | 2,401 | 0.50 | 0.50 | 0 | 1 | 2,433 | 0.51 | 0.50 |
| Age | 4,779 | 56.96 | 14.70 | 20 | 97 | 4,779 | 55.53 | 16.98 |
| Educational attainment | | | | | | | | |
| Low | 846 | 0.18 | 0.38 | 0 | 1 | 859 | 0.18 | 0.39 |
| Intermediate | 1,579 | 0.33 | 0.47 | 0 | 1 | 1,442 | 0.30 | 0.46 |
| High | 2,312 | 0.49 | 0.49 | 0 | 1 | 2,436 | 0.51 | 0.50 |

**Table 2** Sample composition of the GESIS Panel wave jb (n = 4786).

*Note:* For the values of the sociodemographic variables of the panelists information from the recruitment survey was used.

Panel Association (https://openpanelalliance.org), especially in close cooperation with the Dutch LISS Panel (https://www.lissdata.nl), the Understanding American Study (https://uasdata.usc.edu), the Department of Economics of the University of Bonn as well as the German Research Institute of the Federal Employment Agency. That is, here lie additional comparative research opportunities.

In the following[5], we will focus on some multi-item scales that we deem interesting for the audience of this journal, and which measure people's responses to the COVID-19 pandemic: (a) "Assessment risk of infection"; (b) "Adopted measures to prevent infection"; (c) "Evaluation of measures"; and (d) "Trust in politics and institutions". For a detailed description of the measures, see the GP study description for wave jb (GESIS, 2023c). All items can be found in the GESIS Panel Codebook, which also includes information on the most recent wave jb (Weyandt, 2023).

*Assessment risk of infection*: The scale on the "assessment risk of infection" with the Coronavirus comprised five questions about perceived infection risks and attendant consequences: (a) the respondent's perceived personal risk (i.e., probability) of contracting the virus; and (b) the respondent-rated risk of close contacts (friends, family, colleagues); (c) the respondent's personal risk of hospitalization due to Coronavirus; (d) having to quarantine in the next two months, and (e) infecting others. Each item was to be answered on a seven-point rating scale ranging from 1 (not at all likely) to 7 (absolutely likely).

*Adopted measures to prevent infection*: This scale comprised ten dichotomous (0: no, 1: yes) items such as "avoided certain (busy) places," "washed my hands more often and longer," and "stocked up on water and/or food supplies."

*Evaluation of measures:* The scale on beliefs in the effectiveness of policy measures (labeled "Evaluation of measures" in the GP codebook) comprised seven items addressing measures such as the closure of "day-care centers, kindergartens, and schools," "sports clubs and fitness centers," "bars, cafes, and restaurants," and "all shops except supermarkets and pharmacies." Items were to be answered on a five-point rating scale ranging from 1 (not effective at all) to 5 (very effective).

*Trust in politics and institutions*: Finally, the scale on trust in policymakers and institutions comprised relevant actors and institutions dealing with COVID-19. All in all, nine different actors/institutions were presented, such as "your primary care physician," "the local health authority", "the Federal Government," "the Ministry of Health", and "the World Health Organization." The items were to be answered on a five-point rating scale ranging from 1 (don't trust at all) to 5 (trust completely).

Since these (and other) pandemic-related instruments are part of the cumulative dataset, a wide range of background information is available. That is, we provide information on standard socio-demographic variables (gender, age, education, employment), which are updated once a year. In addition, we provide relevant psychological instruments such as the Short Big Five Inventory-2 (German Version of BFI-2-S Danner et al., 2019; BFI-2-S, Soto & John, 2017). The BFI-2-S was fielded in mid-2017 (Wave ec, June-July 2017). The BFI-10 (Rammstedt & John, 2007), as well as the Schwartz value scale (Schwartz & Boehnke, 2004), are measured on an annual basis. Additional relevant (and annually updated) scales are related to subjective well-being, social and political participation, environmental attitudes and behavior, media usage, work and leisure, or attitudes toward refugees.

## 2.4 QUALITY CONTROL

The GP is an open data collection infrastructure that invites researchers from various disciplines to submit survey items and instruments to be fielded. The submission process encompasses two review stages. The first stage focuses on methodological aspects of a submission, including the validity of the survey instrument(s). Once a submission has successfully passed the first stage, the GP team invites two or three external reviewers to conduct a peer review, which also ensures a sound theoretical foundation of a study submission.

During the data preparation process of each GP wave, the data undergo several steps of validation. Any coding of missing data is standardized and guided by the GP missing value scheme. The GP disseminates the data in a rather raw format. This means the data are not validated with respect to previous statements of the panelists within the questionnaire, e.g., filter questions. Usually, this becomes apparent in the offline mode, where panelists can ignore routing and fill in filtered questions anyway. Since the GP is a self-completed survey, issues with inconsistent data can also not be avoided completely, e.g., differences in demographic variables. The latter issue, however, can be easily identified by utilizing the GP demography dataset, which compiles this information across waves, and any inconsistencies are highlighted by flag variables (Kolb et al., 2022, see Section 2.3).

## 2.5 DATA ANONYMIZATION AND ETHICAL ISSUES

The GP as a permanent longitudinal survey infrastructure started in 2013. At that time, there was no need for a formal review from an institutional review board. Nevertheless, the GP is guided by the current ethical and data protection regulations. Thus, during recruitment for the panel (and all subsequent refreshments), the selected individuals are informed in detail about the character (repeated surveys) and the objectives of the study, the voluntary nature of participation, the origin of their personal data, the handling of survey data, and

their rights with respect to data privacy (e.g., right to data deletion). This is also done in each survey wave in the form of an enclosed data protection leaflet. The externally submitted studies are also checked to determine whether the topics or questions are considered particularly sensitive or too emotionally stressful.

With respect to data privacy, specific variables undergo an anonymization process and are either categorized or are only accessible in the so-called GP Extended Edition. This is primarily the case for demographic variables but also relates to sensitive topics as well as spatial information. The panel itself is subject to cleaning processes. Panelists are removed from the panel after three subsequent waves of non-response or non-contact (Minderop et al., 2023).

## 2.6 EXISTING USE OF DATA

The full GP bibliography can be found here and includes about 200 entries: https://search.gesis.org/?type=publication&data_source=GESIS%20Panel-Bibliography.

Concerning our COVID-19-related data collection, we would like to highlight the following papers by Lüdecke and von dem Knesebeck (2020), Schaurer & Weiß (2020), Rammstedt et al. (2022), or Stier et al. (2022).

# (3) DATASET DESCRIPTION AND ACCESS

The GP data – including the GP Corona special survey – is being hosted at the GESIS Data Archive. All in all, we offer three[6] data versions: (1) The "GESIS Panel Standard Edition" is the scientific use file (SUF). It is a cumulative

data set that includes data from all waves, including all COVID-19-pandemic-related items. (2) The "GESIS Panel Extended Edition" is a secure scientific use file for on-site use at the GESIS Secure Data Center in Cologne or Mannheim;[7] it is also a cumulative data set including data from all waves plus sensitive variables. (3) The "Special Survey on the Coronavirus SARS-CoV-2 Outbreak", which has been published as a public use file (PUF) and can be downloaded without any restrictions. It is limited to the corona-related survey fielded in March 2020, which only covers the online subsample. In terms of research opportunities, it has been superseded by the two aforementioned versions.

## 3.1 REPOSITORY LOCATION

The GP is a longitudinal survey infrastructure that, so far, has published 48 cumulative datasets. All data can be downloaded from the GESIS data archive, which provides long-term (at least 25 years) storage of the GESIS Panel data. However, potential users need to register with GESIS before they can download any data. Registration at GESIS is free of charge and open to all. Once you have a registered account, GESIS offers data at different access categories, which are usually determined by the data holder. See Table 3 for more information on access categories, the DOI of the most recent data set as well as links to future data versions.

## 3.2 OBJECT/FILE NAME

The most recent data release, Version 48.0 (wave jb), contains at the root level two folders (`data` and `documentation`) and two files (`Benutzungsordnung_GESIS_DAS.pdf` and `readme.txt`). The `data` folder contains three subfolders for

| EDITION | CATEGORY | DOI/FUTURE-PROOF URL |
|---|---|---|
| GESIS Panel – Standard Edition (GESIS, 2023b) | C – Data and documents are only released for academic research and teaching after the data depositor's written authorization. For this purpose the Data Archive obtains a written permission with specification of the user and the analysis intention. | https://doi.org/10.4232/1.14111 https://search.gesis.org/research_data/ZA5665 |
| GESIS Panel – Extended Edition (GESIS, 2023a) | C – Data and documents are only released for academic research and teaching after the data depositor's written authorization. For this purpose the Data Archive obtains a written permission with specification of the user and the analysis intention. For data protection reasons, access to the data is only granted On-Site in the Secure Data Center at GESIS in Cologne or Mannheim. | https://doi.org/10.4232/1.14110 https://search.gesis.org/research_data/ZA5664 |
| GESIS Panel Special Survey on the Coronavirus SARS-CoV-2 Outbreak in Germany | A – Data and documents are released for academic research and teaching. | https://doi.org/10.4232/1.13520 |

**Table 3** GESIS Panel data editions and access categories (DOI refers to data version 48-0-0, wave jb).

three data formats (`csv`, `spss`, and `stata`). The `documentation` folder provides additional subfolders for `codebook-and-questionnaires`, `reports`, and `study-descriptions`. In addition, we also provide a "cheatsheet" that offers a quick overview about the dataset, a data manual as well as a version history, which also includes errata information. All in all, we provide 190 files in 9 folders.

Since we publish a cumulative dataset, this version also includes the initial "GESIS Panel Special Survey on the Coronavirus SARS-CoV-2 Outbreak in Germany". Most users are advised to use the most recent GESIS Panel Standard Edition of the dataset, which is much more informative than the cross-sectional Special Survey dataset. The only advantage of the Special Survey data is its lower access category ("Data and documents are released for everybody"), which does not require users to complete a "Data User Agreement" and might therefore be suitable for teaching purposes.

Each[8] data release is accompanied by an extensive readme.txt file, which provides some metadata as well as a comprehensive description of all folders and files. Listing 1 provides an extract of readme.txt.

### 3.3 DATA TYPE
The (cumulative) datasets contain information from the self-administered surveys based on online or paper questionnaires. In addition, and depending on the respective survey mode, paradata has been added to the datasets (a detailed description of the available paradata can be found in Weyandt et al., 2022).

### 3.4 FORMAT NAMES AND VERSIONS
All data is provided in three formats. First, we offer semicolon-delimited CSV files, which is an open format and can be used with any statistical software package. We also provide SPSS and Stata datasets.

It should be noted that we cannot provide the cumulative dataset as one file, which would be rather large and might, therefore, not work with some computers. Instead, we divided the dataset into multiple chunks. Each chunk represents one cohort (either the initial recruitment or refreshment cohorts) and all waves of one year. Cohorts are abbreviated with a letter that signals the year they were recruited. "A" stands for the year 2013, the year of the first recruitment for the GP. The waves are abbreviated with two letters: The first letter again signals the year, while the second letter represents the number of the wave in the corresponding year. For instance, the Stata data file "ZA5664_a1_ba-bf_v48-0-0.dta" refers to the initial cohort (a1) and to waves ba to bf (i.e., the first to the sixth wave in 2014). On the other hand, file "ZA5664_f1_ia-id_v48-0-0.dta" refers to the second refreshment cohort (f1, i.e., recruited in 2018), covering the waves ia to id (i.e., all four survey waves in 2021).

However, in order to spare users, who are interested in using the entire cumulative dataset, to manually combine all chunks, we offer R and Stata code files `ZA5664-65_merge-and-append-files.{R, do}` that compile all chunks into one dataset. Moreover, the GP Cheatsheet (https://www.gesis.org/fileadmin/upload/GESIS_Panel/Cheatsheet/gesis_panel_cheatsheet_01.pdf) provides

```
******************************************************************************************
*
*        GESIS Panel Data Release 48-0-0 (2023-06-15)
*
*        GESIS Panel Extended Edition:    doi:10.4232/1.14110
*        GESIS Panel Standard Edition:    doi:10.4232/1.14111
*
*        The following document points at the relative location of the documentation and data for
*        the scientific use file of the GESIS Panel Standard Edition for the above mentioned data
*        release.
*
*        The data are available as cohort specific master files (Stata and SPSS), including all waves,
*        as well as yearly files including all waves fielded for the respective cohort.
*            Stata Master file Cohort 2013    ./data/stata/ZA5665_a1_v48-0-0.dta
*            Stata Master file Cohort 2016    ./data/stata/ZA5665_d1_v48-0-0.dta
*            Stata Master file Cohort 2018    ./data/stata/ZA5665_f1_v48-0-0.dta
*            Stata Master file Cohort 2021    ./data/stata/ZA5665_i1_v48-0-0.dta
*            Stata Demography dataset                  ./data/stata/ZA5665_demography_v48-0-0.dta
*
*            SPSS Master file Cohort 2013    ./data/spss/ZA5665_a1_v48-0-0.sav
*            SPSS Master file Cohort 2016    ./data/spss/ZA5665_d1_v48-0-0.sav
*            SPSS Master file Cohort 2018    ./data/spss/ZA5665_f1_v48-0-0.sav
*            SPSS Master file Cohort 2021    ./data/spss/ZA5665_i1_v48-0-0.sav
*            SPSS Demography dataset                   ./data/spss/ZA5665_demography_v48-0-0.sav
*
*        For more detail on the data release take a look at the Data Manual:
*            ./documentation/ZA5664-65_sd_data-manual.pdf
*
*        All documentation documents can be found here:
*            ./documentation/
*
******************************************************************************************
```

**Listing 1** Extract of readme.txt, which describes the content of the most recent data release.

an "at a glance" overview of the data structure, the missing value scheme, the naming convention for the waves and variables, the fielded core studies, and the different cohorts included in the GP.

### 3.5 LANGUAGE

While the survey was conducted in German, documentation (wave report, codebook, study description, data manual) is in English. The dataset, though, contains German variable labels and value labels.

### 3.6 LICENSE

The GP Standard, as well as the Extended Edition, are only released for academic research and teaching after the data depositor's written authorization. For this purpose, the GESIS Data Archive obtains a written permission with specification of the user and the analysis intention. The GP Corona, however, can be accessed by everybody (i.e., Access Category 0: "Data and documents are released for everybody"). More information about GESIS' terms of use can be found on the following website: https://www.gesis.org/en/data-services/about-the-data-services/standards-and-workflows-data-services/data-access-access (accessed on 2023-08-17).

### 3.7 LIMITS TO SHARING

The GP data is not under embargo. As soon as a wave's data preparation is completed, it will be published immediately. Data requests, however, will be rejected when they fail to provide information about the scientific use of the data or the proper means to ensure the security of the dataset on a user's computer.

### 3.8 PUBLICATION DATE

Publication date of wave jb data (Version 48-0-0): June 15, 2023.

### 3.9 FAIR DATA/CODEBOOK

We believe that, to the best of our knowledge, all GP data versions adhere to the FAIR guidelines. More information is provided by the GESIS Department of Data Services for the Social Sciences (https://www.gesis.org/en/institute/departments/data-services-for-the-social-sciences, accessed on 2023-02-17) and the GP website of the data archive (https://search.gesis.org/research_data/ZA5665, accessed on 2023-02-17).

## (4) REUSE POTENTIAL

### 4.1 OVERVIEW

Since the GP is a large-scale, population-representative, multi-topic, longitudinal study that has been covering the pandemic from its start, the reuse potential is manifold. In fact, part of the mission statement of the GP is to facilitate secondary research by providing a

very well-maintained dataset as well as high-quality documentation, which includes a data manual, a wave report, a codebook, as well as a study description.

First, at the time of writing this article, wave jb (2022) has been published, including ten waves (and the special survey) that address pandemic-related issues. However, the data collection has been continued, and for some constructs, we will provide more than a dozen measurement points for various instruments allowing for sophisticated longitudinal analyses. This, among other, includes topics such as the assessment of the risk of infection, adopted measures to prevent infection, or trust in politics and institutions in dealing with the pandemic (see Table 1).

Second, beginning in early 2021, we also started covering vaccination issues, e.g., vaccination intention or vaccination concerns. The issue of vaccination is significant in several respects, as it is the most effective method for containing severe pandemic consequences, but at the same time, it is highly controversial among some parts of the population. The GP not only enables analyses of the course of attitudes and behavior towards vaccination at the individual level. Based on its numerous background variables, the GP also allows drawing a clear profile of both those people who are still skeptical about vaccination against COVID-19 today, but also of those who have decided to vaccinate after initial hesitation.

Third, the GP has a long tradition of providing psychological research with relevant constructs (some of them have been introduced in Section 2.5) that, for instance, can be utilized as antecedents of pandemic-related outcomes. Since the GP is an open infrastructure, mostly profiting from user submissions, we also provide more "exotic" measures and instruments such as "Implicit theories about willpower" (study code: bt), "Envy in Daily Life" (study code: ba) or "Dispositional self-compassion" (study code: cb) (the respective study codes and descriptions can be found in GESIS, 2023c).

Fourth, since the GP data also contains geographical information such as federal state, aggregate information at the state level can be linked to the data. For instance, the "The ZPID Lockdown Measures Dataset for Germany" contains information on multiple public health measures and spans from moderate recommendations such as physical distancing, up to the closures of shops and bans of gatherings and demonstrations (Steinmetz et al., 2020). More recent small-area data can be found at "Corona Daten Deutschland", a project of the "Statistisches Bundesamt" (https://www.corona-daten-deutschland.de/, accessed on 2023-02-17). Here, researchers can find data on governmental measures, excess mortality, vaccination, intensive care units, employment measures, etc. Please note, however, that linking geographical information to the GP and analyzing this linked data requires working within the secure environment of the Secure Data Center in Cologne or Mannheim.

Fifth, the data may help to inform public health campaigns by identifying groups at high risk of being infected. While clinical studies allow for examining medical correlates of the susceptibility to an infection, GP data may supplement the drawing of such risk profiles with its wide range of demographic, psychological, and behavioral variables. Moreover, when it comes to vaccination, clinical studies on COVID-19 mostly include volunteers holding particularly positive attitudes toward vaccination. This selection bias is far less pronounced in the GP as it is a multi-topic survey without a particular focus on COVID-19. This variability in attitudes towards vaccination, combined with the many characteristics that can be used as independent variables, makes the GP data highly attractive for identifying the antecedents of skepticism towards COVID-19 vaccinations.

## 4.2. CASE STUDY

In the following, we present a longitudinal use case for psychological research using data on the reported intention to vaccinate against COVID-19. More specifically, we draw on the vaccination item at two measurement points and predict the individual trajectories of stability and change with the value dimension "openness to change" from the Schwartz values inventory (Schwartz, 2012), annually surveyed in the GP.

In Germany, the vaccination campaign against COVID-19 started early in 2021 after the first vaccine received approval in December 2020. Accordingly, GP participants were asked in the first survey wave in 2021 (wave ia) about their intention to be vaccinated. Respondents could answer this question on a scale ranging from 1 ("Definitely not") to 7 ("Definitely"). Additionally, the answer option "I am already vaccinated" was offered. This question was asked in the same way in all subsequent survey waves.

Regarding the Schwartz-Values, we rely on their measures in wave hd, i.e., three survey waves before wave ia, which was fielded between August and October 2020. For measuring the dimension "openness to change", the GP relies on five items, namely:

- It is important to her/him to always form her/his own opinion.
- It is important to her/him to expand her/his knowledge.
- It is important to her/him to have a variety of new experiences.
- It is important to her/him that she/he has the freedom to choose what she/he does.
- It is important to her/him that she/he gets to the bottom of things herself/himself and understands them.

For each of these items, respondents are asked to rate how similar, or dissimilar the person described is to them on a scale from 1 ("not like me at all") to 6 ("very much like me").

In the following, we rely on the responses of those 3,788 individuals who answered the five items measuring the "openness to change" dimension in wave hd and also gave valid responses with respect to the question on vaccination intention in the first and last survey wave in 2021 (waves ia and id, fielded between February and April 2021, and between November 2021 and January 2022, respectively).

For our analytic sample, the five items measuring openness to change show acceptable reliability ($\alpha = .77$, $\omega = .77$). Thus, we created a summative index ranging from 0 to 1, with higher values indicating more openness to change (M = .77; SD = .13). Regarding the vaccination item, 4.88% of the respondents stated to have already been vaccinated in wave ia. In addition, 59.76% responded that they would get vaccinated in any case. In wave id, the respective proportions were 85.04% (already vaccinated) and 6.60% (getting vaccinated in any case). On the contrary, 15.75% in wave ia answered that they were unlikely or definitely not going to get vaccinated (scale points 1–3). This proportion reduced to 6.83% in survey wave id. Overall, these results clearly suggest that respondents' willingness to be vaccinated increased significantly between the two survey waves.

How does individual openness to change influence stability and change with respect to vaccination intentions? To investigate this, we first grouped together those respondents who indicated that they either had already been vaccinated or that they intended to be vaccinated in any case. In this way, we take into account the different opportunity structures (i.e., differences in the availability of vaccines) at the two points of measurement. Second, we checked whether respondents differed in their intention to vaccinate against COVID-19 initially (i.e., in wave ia) by their degree of openness to change. According to Spearman's rho, this is not the case ($\rho = .002$; $p = .911$). Hence, differences in the trajectories of vaccination intent between respondents who differ in their reported openness to change cannot be attributed to different base levels of vaccination intent. Finally, we created a new variable indicating whether respondents' intention to vaccinate remained the same (base outcome) and whether it increased or whether it decreased. This new group variable served as the dependent variable in a multinomial logistic regression model, with openness to change as the sole predictor.

Compared with the base outcome (stability), a higher openness to change reduces both the likelihood of increased ($b = -2.19$; $p < .01$) and decreased ($b = -0.68$; $p < .05$) vaccination intent. This is also illustrated in Figure 1, in which the predicted probabilities for the three outcomes are plotted for varying levels of openness to change.

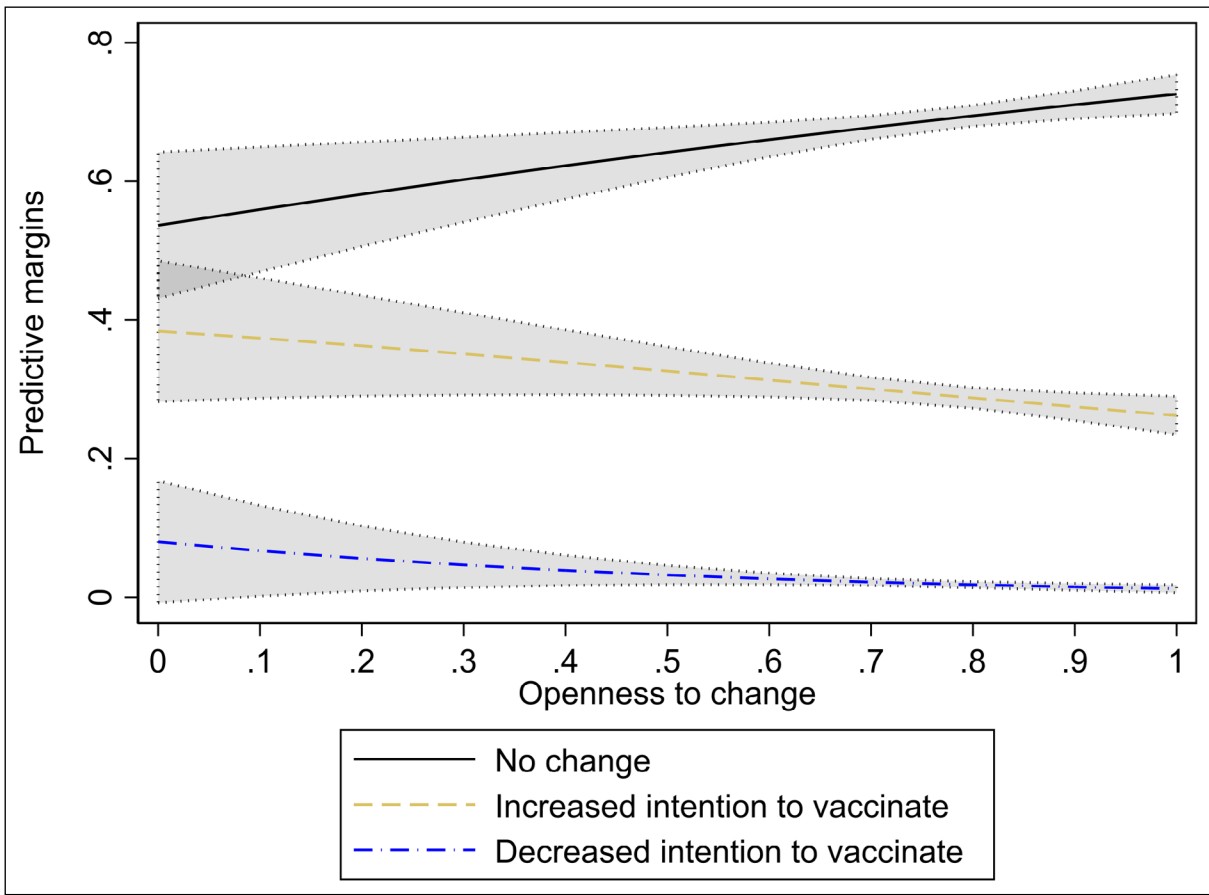

**Figure 1** Predicted probabilities of stability and change of vaccination intent with varying openness to change (grey area: 95% CI).

For respondents with comparatively low values for openness to change (.6, i.e., the 10th percentile), the model predicts a probability of stability of 66.02%, while for respondents with relatively high values for openness to change (.92, i.e., the 90th percentile) the respective predicted probability is 71.36%.

The finding that higher values for openness to change *decrease* the probability of a change in vaccination intention may be surprising at first glance. It is conceivable, however, that this is due to a correlation between openness to change and formal education ($\rho$ = .013; p < .001), with more educated respondents being firmer in their beliefs (Xu et al., 2020). Yet, if formal education is introduced into the model, the effects of openness to change hardly change.

In sum, we hope that these brief analyses have demonstrated that the GP data have much potential when it comes to investigating the influence of personality traits and value orientations on pandemic-related attitudes and behaviors.

## NOTES

1   Since 2013, the GESIS Panel has offered the scientific community the opportunity to collect survey data (mostly) free of charge within a probability-based mixed-mode (mail and online) panel (Bosnjak et al., 2018; Weiß et al., 2020). See Section (2) for more details.

2   On the GP website, the Documentation section (https://www.gesis.org/en/gesis-panel/documentation) contains some of the documents referenced here, such as the Wave Reports, Codebook, Study Descriptions, and Data Manual.

3   More information about the study submission process can be found on the following website: https://www.gesis.org/en/gesis-panel/submission (accessed on 2023-03-10).

4   The Federal Statistical Office (Statistisches Bundesamt) figure for the highest upper secondary leaving certificate includes both the *Fachhochschule* (University of Applied Sciences) and the university entrance qualification.

5   This section makes use of Rammstedt et al. (2022).

6   We also offer the so-called „GESIS Panel Campus File", which is a PUF (Public Use File, which has no restrictions in terms of access) and can be used for teaching purposes. It is a 25% sample of the six waves from the year 2014.

7   Access via a remote desktop approach is currently under development and expected to be available in the near future.

8   The exception being the GESIS Panel Special Survey on the Coronavirus SARS-CoV-2 Outbreak in Germany, which, however, provides a comprehensive codebook.

## ACKNOWLEDGEMENTS

Initially, the module "cy: Coronavirus Outbreak in Germany" was developed and implemented by Bernd Weiß, Ines Schaurer, and Mirjan Kummerow. These developments were embedded in larger ad-hoc efforts to measure pandemic-related outcomes that included the LISS Panel, the Institute for Employment Research (IAB), Bonn University, the Johns Hopkins Bloomberg School of Public Health, as well as the GESIS Panel team.

Later versions of module "cy" have benefited from inputs from Isabella Minderop, Maikel Schwerdtfeger, and Sven Stadtmüller. The Open Access Fund of the Leibniz Association funded the publication of this article.

## FUNDING INFORMATION

All data collection is founded by GESIS – Leibniz Institute for the Social Sciences. GESIS is founded by federal and state governments.

## COMPETING INTERESTS

The authors have no competing interests to declare.

## AUTHOR AFFILIATIONS

**Bernd Weiß** ⓘ orcid.org/0000-0002-1176-8408
GESIS – Leibniz Institute for the Social Sciences, Germany

**Sven Stadtmüller** ⓘ orcid.org/0000-0002-2369-002X
GESIS – Leibniz Institute for the Social Sciences, and Frankfurt University of Applied Sciences, Germany

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

## PEER REVIEW COMMENTS

*Journal of Open Psychology Data* has blind peer review, which is unblinded upon article acceptance. The editorial history of this article can be downloaded here:

- **PR File 1.** Peer Review History. DOI: https://doi.org/10.5334/jopd.90.pr1

**TO CITE THIS ARTICLE:**
Weiß, B., & Stadtmüller, S. (2023). Using the Probability-Based GESIS Panel for Longitudinal Psychological Research on the COVID-19 Outbreak in Germany. *Journal of Open Psychology Data,* 11: 16, pp. 1–12. DOI: https://doi.org/10.5334/jopd.90

**Submitted:** 17 March 2023    **Accepted:** 07 September 2023    **Published:** 29 September 2023

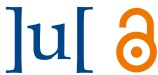