## [Peer Review History. · Journal of Open Psychology Data]

Dear Editors,

Dear Reviewers,

We hereby submit a revised version of our article "Using the probability-based GESIS Panel for longitudinal psychological research on the COVID-19 outbreak in Germany". Thank you very much for your helpful suggestions, which we implemented in our revision of the paper. We believe that addressing the comments has helped us to improve the overall quality of the paper.

Please find below a point-by-point response to all concerns that were raised by the reviewers. We hope that you find our responses satisfactory and that the manuscript is now acceptable for publication.

We also have added the Stata code that have been requested by one of the reviewers. We would appreciate if the code could be published as well.

Thank you very much.

Best wishes

Bernd Weiß and Sven Stadtmüller

We thank reviewer 1 for their extensive, detailed and helpful suggested edits and comments. We have accepted and incorporated almost all of their suggestions. Please find our responses in red. We also would like to mention that we have updated our manuscript to the most recent dataset that has been published (wave “jb” instead of “id”, i.e., two additional waves)

Review 1

The submission titled “Using the probability-based GESIS Panel for longitudinal psychological research on the COVID-19 outbreak in Germany” presents a very impressive infrastructure including a large longitudinal dataset focused on the COVID-19 pandemic. It has a great potential for future reuse not only because of the important topic of the project but also because of the meticulous data management and documentation. I would like to congratulate the authors for coordinating such great efforts. Below, I provide feedback based on the manuscript content and data requirements for the journal. Further, I also list recommendations for clarifying some elements of the manuscript.

Journal Requirements

Paper Contents:

1. Sufficient detail about the data: Yes. The data belong to a complex infrastructure and the current manuscript explains selected elements of that infrastructure. However, vast documentation on how the data were created can be accessed from the project website.
2. The data must be correctly described: Yes. Considering the complexity and the size of the data with 11 folders and 166 files, this is very difficult to assess. The open access Special Survey data are described correctly and clearly. Assuming that a similar approach has been taken with the rest of the data, I believe that this requirement is also met.

We have updated the number of folders and files in the manuscript.

3. Concrete reuse suggestions: Yes. The authors provide a number of interesting points for future reuse of the data and also demonstrate one way that the data could be used for analyses.

The deposited data:

3. Repository sustainability – Authors should clarify. See my comment 3.a below.
4. Open Access – Not all data are open access. See my comments 3.d and 3.e below
5. Open Format – Yes. The data are available as CSV files
6. Appropriate labelling – Yes. Variable labels are complex because the data are complex, but the Variable Naming Convention has been explained in the overall Project Codebook.
7. Actionable data – Yes. No specialised software is needed to access and use the data.
8. Ethics – Authors should clarify. See my comment 2.h below.

Recommendations:

1. *Background*

1. You write “had the entire country “ and of course, I understand that this means Germany but it would aid clarity if this was also specified in this sentence.

*Thank you very much for your comment! Indeed, we have not emphasized the geographical context in the introduction. Therefore, we have updated the very first sentence: “The COVID-19 pandemic had (and still has) an immense impact at the societal and individual level in Germany.” As well as the sentence in question: “Since the pandemic, despite all regional differences, had **entire Germany** firmly in its grip [...]”*

2. *Methods*

2. 2.1. It would help to specify here what is meant by the probability-based design. It is a big strength of the panel so it would be good to draw attention to this early on for potential future users.

Many thanks for this suggestion. We have now elaborated on this aspect at the beginning of chap. 2.1: “The GP is a probability-based survey. This means that the selection of participants is based on a random sample of individuals from the German-speaking residential population.”

2. 2.1. I find the following sentence confusing “Eight of these 1 + 13 waves have already been published, i.e., waves hz to id (see Table 1), and can be used for data analyses.”. First, please provide a reference for the published datasets. Second, the fact that some of the data have been published but not all makes it difficult to understand which parts of the data the current manuscript applies to. Please clarify whether the current manuscript applies to the first eight waves only. If it applies to all, then please add information about how this manuscript applies to the data that will be published in the future and what the future reader might expect when the remaining data are published.

Thank you for this important advice. We revised this paragraph as follows: “Until now (August 2023), 1 special survey and 13 panel waves, including COVID-19 items, were fielded. The special survey (study hz) and ten of these 13 waves have already been published, i.e., the regular survey waves hb to jb (see Table 1). Each new data release contains the cumulative data set from all previous survey waves as well as from GP Corona. The most recent publication that can be used for data analyses is version 48 (GESIS, 2023a; 2023b). Hence, the recommended way to work with all GP COVID-19 instruments is to obtain the most recent GESIS Panel Standard/Extended Edition. Accordingly, we refer to this version when we report on GP data.”

2. 2.1. Please revise the following sentence: “The questionnaire module on the COVID-19 outbreak has been continuously updated for the last three years as the measures to contain the pandemic varied over time.”

We have revised the sentence as follows: Since the measures to contain the pandemic varied over time, we have updated the questionnaire module on the COVID-19 outbreak for the last three years.

1. Table 1 – Provide a more detailed caption for the table. Explain that the lowercase letters in blue boxes represent codes for different waves. Explain the superscript letters in relation to the affiliated groups. Why are these affiliations listed?

The caption now includes more information on the coding scheme of the GESIS Panel waves and on why the other groups were listed.

2. 2.2. Briefly explain what is meant by and how a unified mode approach was implemented in GP. This is a strength of the collected data that will be of interest in future reuse.

Thank you for this important suggestion. We have complemented the paragraph as follows: "To reduce mode measurement effects, the GP pursues a unified mode approach (Dillman & Edwards, 2016). This means that the presentation of questions and response options between the two modes of data collection (i.e., web and mail) is implemented as similarly as possible. To account for the increasing share of panelists taking part with mobile devices, we also converted the questionnaire into a mobile-first layout in 2020. This means that the questions and response options are also presented similarly between the different devices that can be used for web survey participation (i.e., desktop/laptop, tablet, and smartphone devices). Specifically, all questions and scales are now arranged vertically with up to five survey items per page of the web questionnaire. Consequently, the paper questionnaire was also rearranged from a single- to a two-column layout to account for the additional space the new layout required (Schwerdtfeger et al., 2022)."

2. 2.2 This section has repetition with section 2.1., which could be reduced. Example paragraph three in section 2.1. and paragraph two in 2.2.

Thank you very much for this suggestion. We have removed these redundancies.

2. 2.3. I agree that it is a good idea to offer a description of just a selection of measures that focus on COVID-19 specifically. However, I think it's important to provide a full list of administered measures that can be found in the data. If the list is very long, consider adding it as an appendix so that the reader can access it easily.

Since we provide very detailed and well-maintained documentation, we would like to abstain from adding a list of items to the article but, instead, refer interested readers to the GESIS Panel codebook. We, therefore, added the following sentence: All items can be found in the GESIS Panel Codebook, which also includes information on the most recent wave jb (Weyandt, 2023).

2. 2.5. Please specify your procedures for obtaining informed consent from panellists.

Thanks for hinting to this important aspect. We added the following paragraph in chap. 2.5. which deals with these procedures: "Nevertheless, the GP is guided by the current ethical and data protection regulations. Thus, during recruitment for the panel (and all subsequent refreshments), the selected individuals are informed in detail about the character (repeated surveys) and the objectives of the study, the voluntary nature of participation, the origin of their personal data, the handling of survey data, and their rights with respect to data privacy (e.g., right to data deletion). This is also done in each survey wave in the form of an enclosed data protection leaflet. The externally submitted studies are also checked to determine whether the topics or questions are considered particularly sensitive or too emotionally stressful."

3. Data Description and Access

3. 3.1. Please comment on the longevity and sustainability of this archive.

*The GESIS Panel data are archived by GESIS – Leibniz Institute for the Social Sciences, which is Europe's largest infrastructure institute for the social sciences. GESIS also offers long-term data archiving services. Therefore, we have updated the respective description: "All data can be downloaded from the GESIS data archive, **which provides long-term storage (at least 25 years) of the GESIS Panel data.**"*

3. 3.2 I understand that this section talks about the Standard Edition of the data. This should be specified. Besides, it appears to me that the Special Survey Data, which is openly available to all, had no readme.txt file. There is, however, a very comprehensive Codebook, which explains the data. This should be clarified here.

Thank you very much for your comment. We have added a sentence suggesting future users to use the Standard Edition instead of the Special Survey. We also explain that the Special Survey does not require a Data User Agreement, which might be considered an advantage. Finally, we have added a new footnote explaining that the Special Survey does not include a readme.txt file.

3. 3.4. As far as I can see, the CSV files are semicolon separated rather than comma separated. Please specify this in the manuscript.

*We have added the following description: "First, we offer **semicolon-delimited** CSV files, [...]"*

3. 3.6. Please also specify the licence of the open data set for the Special Survey

Thank you very much for your advice. We have added the following sentences: "The GESIS Panel Special Survey, however, can be accessed by everybody (i.e., Access Category 0: "Data and documents are released for everybody"). More information about GESIS' terms of use can be found on the following website: <https://www.gesis.org/en/data-services/about-the-data-services/standards-and-workflows-data-services/data-access-access> (accessed on: 2023-08-17)."

3. 3.7. Since the data are not fully open, please specify any reasons why a request for the use of the data could be rejected.

We have added the following explanation: "Data requests will be rejected when they fail to provide information about the scientific use of the data or the proper means to ensure the security of the dataset on a user's computer."

4. *Reuse potential*

4. 4.1. "This variability in attitudes towards vaccination, combined with the many characteristics that can be used as independent variables, makes the GP highly data attractive for identifying the antecedents of skepticism towards COVID-19 vaccinations." I think that the word order for "highly data" should be reversed here.

We have reversed the order of the two words.

4. 4.2. Thank you for providing the case study. This is a good demonstration of what could be done with the data. Is it possible that authors could share the script that generated these results?

We are happy to share the Stata Dofile, which we will provide with the revision and ask the editors to upload it as an additional file.

5. *Other*

1. There are some references that were difficult to find because they have no DOI. These include GESIS (2022d), Kolb et al. (2022), Stadtmüller et al. (2022) and Weyandt et al. (2022). I have found them all here <https://www.gesis.org/en/gesis-panel/documentation> This website is not listed anywhere in the manuscript. I suggest it should be either added to the main text or linked in the references.

Thank you for this valuable advice. We added footnote 2 at the very beginning of our manuscript in which we provide the link to our documentation materials. Here, we also indicate that some of the documents we refer to throughout the manuscript can be accessed there.